# Z-Scheme Heterojunction of 3-Dimensional Hierarchical Bi_3_O_4_Cl/Bi_5_O_7_I for a Significant Enhancement in the Photocatalytic Degradation of Organic Pollutants (RhB and BPA)

**DOI:** 10.3390/nano12050767

**Published:** 2022-02-24

**Authors:** Syed Taj Ud Din, Hankyu Lee, Woochul Yang

**Affiliations:** Department of Physics, Dongguk University, Seoul 04620, Korea; tajuddins.phy@gmail.com (S.T.U.D.); lhk0022@naver.com (H.L.)

**Keywords:** 3D-hierarchical morphology, Z-scheme Bi_3_O_4_Cl/Bi_5_O_7_I heterojunction, photocatalytic degradation, organic pollutant, visible-light absorption

## Abstract

In this study, we report the synthesis of a 3-dimensional (3D) hierarchical Bi_3_O_4_Cl/Bi_5_O_7_I (BOC/BOI) heterostructure for the photocatalytic degradation of Rhodamine-B (RhB) dye and colorless Bisphenol-A (BPA) pollutant under visible light. The heterostructure was prepared using in situ solvothermal and calcination methods. BOC/BOI exhibits a 3D hierarchical structure constructed with thin nano-platelets. The photocatalytic performance of the BOC/BOI photocatalyst demonstrated that the degradation efficiencies of RhB and BPA were 97% and 92% after light illumination within 90 and 30 min, respectively. In comparison, bare BOC and BOI efficiencies were only 20% and 10% for RhB dye, respectively, and 2.3% and 37% for BPA aqueous pollutants, respectively. Moreover, radical trapping measurements indicated that ^•^O_2_^−^ and ^•^OH radicals played prominent roles in RhB and BPA degradation into mineralization. Analysis of band structures and photochemical redox reactions of BOC/BOI revealed a Z-scheme charge transfer between BOC and BOI by an internal electric field formed at the interface. Therefore, the highly improved photocatalytic performance of the BOC/BOI heterostructure is attributed to the synergetic effects of large surface area, high visible-light absorption, and the enhanced separation and transport of photo-excited electron–hole pairs induced by the hierarchical and Z-scheme heterojunction of the BOC/BOI.

## 1. Introduction

As water pollutants have become an increasingly severe threat to life in this industrialized society, removing hazardous pollutants is currently a major social challenge. The pollutants mostly included in water are hazardous organic dyes and phenolic compounds. Rhodamine-B (RhB) is a common water-soluble organic dye used widely in various industry fields. Since it is classified as carcinogenic and neurotoxic, it poses health risks to humans [1]. Thus, its maximum allowable concentration is in the range from 14 to 140 μg/L in water [2]. Other than dyes, the colorless Bisphenol-A is the other frequently used compound. It is also a widely used raw material for the fabrication of numerous polymeric materials [3]. Its no-effect concentration is required to be less than 0.06 μg/L because its long-term exposure causes endocrine-based, neurological, and reproductive developmental disorders [4]. Therefore, the removal of RhB and BPA from the wastewater is crucial before discharging into water reservoirs and landfills. In general, adsorption through microbial treatment and chemical redox reaction has been used to remove these pollutants. However, these approaches have some limitations, due to low efficiency, high cost, and incomplete removal of contaminants [5]. Therefore, the effective elimination of these hazardous organic pollutants from wastewater remains a challenge.

To solve these issues, photocatalysis has attracted substantial attention due to environmental friendliness and its use of freely available solar light [6,7,8]. The photocatalytic process of a semiconductor consists of the absorption of photons, the separation and transport of photo-excited charge carriers, and the redox reaction on the surface. Although various semiconducting photocatalysts with high photocatalytic performance have been reported, there are still some limitations, such as high electron–hole recombination, inappropriate redox potentials, and low capability of visible-light absorption [9]. In particular, to enhance the separation and transport of photo-excited charge carriers, which is a key process during the photocatalytic process, various strategies have been explored including nonmetal doping [10], the deposition of noble or non-noble metal as a cocatalyst on the semiconducting photocatalyst [11,12,13,14], type-II heterojunction [15], and Z-scheme heterojunction photocatalyst [16] through the coupling of two semiconducting materials.

The Z-scheme heterojunction photocatalyst has attracted extensive attention due to the direct transfer and spatial separation of photogenerated charge carriers between two photocatalysts, and maintaining its high redox ability of both surfaces [17]. Various nanocomposites or heterojunctions, such as MoS_2_/Bi_2_WO_6_, g-C_3_N_4_/WO_3_, CdS/Bi_3_O_4_Cl, etc. [13,14,15], have been used to synthesize Z-scheme photocatalysts for the degradation of aqueous organic pollutants. Recently, bismuth oxyhalides (BiOX (X = Cl, Br, I))-based nanocomposites are considered to be promising candidates for use as Z-scheme heterojunction photocatalysts. The layered crystal structure of BiOX composed of the [X]-[BiO]-[X] gives it unique electronic and structural properties, such as bandgap tunability [18], easy adjustment of conduction, and valence band potentials via the different compositions [19], formation of internal electric field between interlayers [20], and possible growth in multiple dimensions [21]. In particular, Bi_3_O_4_Cl (BOC) and Bi_5_O_7_I (BOI) are photochemically stable and have suitable bandgaps of 2.60 eV and 3.01 eV for visible-light absorption, respectively [22,23]. Furthermore, the more negative conduction band (CB) potential of BOI and the more positive valence band (VB) potential of BOC are ideal for the maximum generation of super-oxide (^•^O_2_^−^) and hydroxyl (^•^OH) radicals, respectively, which are beneficial for the degradation of organic pollutants [24].

Therefore, various Z-scheme heterostructures of the other group semiconductors combined with BOC or BOI have been developed for photocatalytic applications. For instance, Zhou et al. prepared a series of BOI/MIL-53(Fe) nanocomposites [6]. The well-matched energy levels of BOI with MIL-53(Fe) enabled the construction of a direct Z-scheme heterojunction for the degradation of RhB under visible-light irradiation. Similarly, Jiang et al. used a facial pH control route to develop a BOC/Bi_12_O_17_Cl_2_ Z-scheme composite by the in situ growth of BOC on the surface of Bi_12_O_17_Cl_2_ for high photodegradation efficiency under visible light [25]. In another study, Che et al. prepared a Z-scheme BOC/g-C_3_N_4_ heterostructure to remove antibiotics, dye, and heavy metal from water under visible light [26]. The high photocatalytic performance is attributed to the Z-scheme heterojunction and the strong interfacial functionalization between the BOC nano-flakes and the g-C_3_N_4_ nanosheets interface.

Meanwhile, the morphology of the photocatalyst strongly affects the photocatalytic performance. Therefore, various morphologies, such as zero-dimensional (0D) nanodots [27], 1D nanorods [28], 2D nanosheets [29], and 3D hierarchical [30,31] nanostructures, have attracted significant attention due to their high surface-to-volume ratio, which increases the amount of photoactive sites [32]. Use of the nanostructures with 3D hierarchical morphologies is a promising approach to develop efficient photocatalysts, as this design inhibits self-aggregation of nanostructures due to high surface energies. However, the photocatalytic performances of 3D hierarchical single semiconductors and conventional heterostructures are far from ready for implementation in the photocatalytic degradation of organic pollutants. Therefore, it is necessary to develop a 3D hierarchical Z-scheme heterostructure to benefit from the synergy of its high surface area, high electron–hole separation, and high redox ability.

In this study, we prepared a Z-scheme BOC/BOI heterojunction photocatalyst with a 3D-hierarchical morphology for enhancement in the photocatalytic degradation of RhB and BPA. The heterojunction was fabricated using an in situ two-step solvothermal-calcination method. Morphological characterizations show that BOC/BOI has a 3D-hierarchical structure constructed with thin nano-flakes. Analysis of energy-band structures of BOC/BOI heterojunction proves that the generation of an internal electric field directed from BOI to BOC facilitates Z-scheme charge transfer between BOC and BOI. Thus, the BOC/BOI demonstrated a higher reaction rate and degradation efficiency than the bare BOC and BOI during RhB and BPA degradation. Furthermore, the degradation pathway was suggested by identifying the mineralization and reaction intermediates using liquid chromatography–mass spectroscopy. The excellent performances of the BOC/BOI heterostructure are attributed to more active sites induced by its unique 3D hierarchical morphology and the enhancement of the separation of photogenerated electrons and holes through the Z-scheme transfer process. This research presents that the Z-scheme BOC/BOI heterojunction with a 3D-hierarchical structure is a suitable photocatalyst for the degradation of environmental pollutants.

## 2. Materials and Methods

### 2.1. Chemicals

Bismuth nitrate pentahydrate (Bi(NO_3_)_3_·5H_2_O, 99%), potassium chloride (KCl, 99%), potassium iodide (KI, 99%), ethylene glycol ((CH_2_OH)_2_, 99%), 1-Vinyl-2-Pyrrolidone (PVP) ((C_6_H_9_NO), 99%), Rhodamine-B (CH_31_ClN_2_O_3_, 99%), Bisphenol-A (C_15_H_16_O_2_, 99%), dimethylformamide (DMF), nitro-blue tetrazolium (NBT), terephthalic acid (TA) and 1,4-benzoquinone (BQ) (C_6_H_4_O_2_, 99%) were purchased from Sigma Aldrich Inc. (St. Louis, MO, USA). ethanol (C_2_H_5_OH, 99%) and isopropyl alcohol (IPA) (C_3_H_8_O, 99%) were purchased from DAEJUN Co., Ltd. (Daejun, Korea). All reagents were used without any further purification.

### 2.2. Preparation of BOC/BOI Composites

An in situ two-step solvothermal method was used to prepare the BOC/BOI heterojunction, as shown in Appendix A. In Step (a), Bi(NO_3_)_3_·5H_2_O (4.5 mmol) was dissolved in 15 mL of ethylene glycol (EG). The mixture was then sonicated and stirred vigorously for 30 min. In Step (b), the PVP (10 mmol) was dissolved in 10 mL of EG, which was then sonicated and stirred vigorously for 30 min. In Step (c), the PVP solution (prepared in Step (b)) was added dropwise into the Bi(NO_3_)_3_·5H_2_O solution (prepared in Step (a)) and stirred for 15 min at 400 rpm. Next, in Step (d), various molar ratios of potassium chloride (KCl) and potassium iodide (KI) were dissolved in 10 mL of EG by stirring for 30 min. In Step (e), the KCl:KI solution was added dropwise into the Bi(NO_3_)_3_·5H_2_O solution (prepared in Step (c)) under continuous stirring. In Step (f), the mixture was then transferred to a 50 mL Teflon-lined autoclave and sealed. Next, the autoclave was heated at 160 °C for 12 h and cooled naturally to room temperature. After that, the raw BOC/BiOI composite powder was isolated from the EG solution using centrifugation at a speed of 8000 rpm for 10 min and washed with distilled (DI) water and ethanol. The washed BOC/BiOI powder was then dried overnight at 60 °C and put into alumina crucibles. The powder was calcinated in a muffle furnace at 450 °C for 1 h with a ramp rate of 5 °C/min. As a result, the targeted BOC/BOI composites were prepared. The samples with various molar ratios of 1:1, 1:2, 1:3.3, and 1:4.6 of KCl and KI were synthesized, and these were respectively denoted as BOC/BOI-1, BOC/BOI-2, BOC/BOI-3, and BOC/BOI-4. In addition, for comparison, bare BOC and BOI samples were prepared with a similar method as the one used for BOC/BOI. For the preparation of bare BOC and BOI, KCl and KI were added relative to the molar ratio of Bi in Bi(NO_3_)_3_·5H_2_O solution, such as Bi:Cl with 3:1 and Bi:I with 1:1.

### 2.3. Characterization of Prepared Samples

Structural analysis of the prepared samples was conducted using X-ray powder diffractometer (XRD)(Ultima IV, Rigaku Inc., Tokyo, Japan) with Cu-Kα as an X-ray source (wavelength of 1.5409 Å) operated at a scan rate of 5°/min in the diffraction angle (2θ) range from 10° to 80°. The crystalline phase was characterized through Micro-Raman spectroscopy measurement (XperRAM100, Nanobase Inc., Seoul, Korea), equipped with a green laser source (wavelength of 532 nm and power of 6 mW). The morphology and elemental compositions were examined using a field emission scanning electron microscope (SEM) (JSM-6700F, JEOL Ltd., Tokyo, Japan) equipped with an Energy-dispersive X-ray spectroscope (EDX). A UV-visible spectrophotometer (V-750, Jasco Inc., Tokyo, Japan) equipped with a 60 mm integrating sphere was used to investigate the optical properties, including the diffuse reflectance spectrum and optical bandgap. X-ray photoemission spectroscopy (XPS) (Veresprobe II, ULVAC-PHI Inc., Kanagawa, Japan) with a Monochromatic Al Kα X-ray source was used to examine the chemical composition of samples. The collected XPS data were calibrated with a non-oxygenated carbon peak that originated at 284.8 eV [30,31]. The photoluminescence (PL) spectra of the samples were obtained using fluorescence spectrophotometer (Fp-8600, Jasco Inc., Tokyo, Japan) with an excitation wavelength of 330 nm. The time-resolved photoluminescence (TRPL) decay spectra were obtained using Spectrofluorometer (FS5-TCSPC, Edinburgh Instruments Ltd., Edinburgh, UK) using 375 nm picosecond pulsed laser diode (EPL-375) as an excitation source. The TG/DTA analyzer (STA 6000, Perkin Elmer, Waltham, MA, USA) was used to collect the thermo-gravimetric analysis (TGA) and differential scanning calorimetric (DSC) data of the synthesized samples (Appendix A). The microstructures of the samples were analyzed using high-resolution transmission electron microscopy (HRTEM: Talos F200X TEM, FEI Co., Hillsboro, OR, USA) operated at an accelerating voltage of 200 kV. The Gatan microscopy software (digital micrograph™, version 3.7.4, GATAN Inc., Pleasanton, CA, USA) was used to obtain fast Fourier transform (FFT), inverse fast Fourier transform (IFFT), and TEM line profile from the HRTEM images.

### 2.4. Photocatalytic Activity Measurements

The photocatalytic characteristics of the samples were determined using a visible light source with a 300 W Xenon lamp (1000 W/m^2^) (CEL-HXF300, CEAULIGHT Co., Beijing, China) equipped with a UV-IR cutoff filter (420 nm > λ > 780 nm). RhB dye and BPA colorless pollutants were used to evaluate the degradation efficiency of the synthesized photocatalysts. The pollutant aqueous solution in a double wall jacket beaker (80 cm^2^ surface area) connected with a water chiller was kept perpendicular to the light source at a distance of 30 cm. Each photocatalyst with 10 mg and 40 mg was used to degrade 40 mL of RhB dye (17 ppm) and BPA colorless pollutant (15 ppm), respectively. Initially, the RhB and BPA solution was stirred for 60 min and 30 min, respectively, under the dark condition to attain adsorption-desorption equilibrium of the photocatalysts. Next, after a specific time interval, a 2 mL solution was taken from each solution, which was then centrifuged for 3 min at 3000 rpm to separate the solution from the photocatalyst. The absorbance spectrum of the separated solution was determined using a UV-visible spectrophotometer, which established the rate of degradation to light irradiation time. Radical trapping experiments were conducted to determine the radical’s dominant species involved in the photocatalytic decomposition of RhB. IPA, BQ, and KI of 2 mmol were used as trapping reagents to determine the active species such as ^•^O_2_^−^, ^•^OH radicals and holes, respectively. NBT (0.025 mmol) was used as an indicator to determine the amount of ^•^O_2_^−^ radical. We also employed a liquid chromatography–mass spectrometry (LC–MS) system (Agilent 6460 Triple Quad LC/MS, Agilent technologies Inc., Santa Clara, CA, USA), installed with a column (ZORBAX SB-C18 18 μL, 2.1 × 50 mm^2^) for the degradation pathway and reaction intermediates investigation.

### 2.5. Electrochemical Impedance Spectroscopy Measurements

The electrochemical impedance spectroscopy (EIS) analysis was conducted on a three-electrode workstation (VSP Potentiostat, Biologic, Seyssinet-Pariset, France). The Pt-spiral wire and standard calomel were used as counter and reference electrodes, respectively. An aqueous solution of 0.5 M Na_2_SO_4_ (50 mL) was used as an electrolyte. A clean fluorine-doped tin oxide (FTO) glass was employed as a substrate of the working electrode. The working electrode was prepared from the slurry, which was made by dispersing the photocatalyst (0.1 mg) in DMF (1 mL), then sonicated for an hour. The slurry was coated over the FTO by a drop-casting technique and dried at 160 °C for 1 h before being used as a working electrode.

## 3. Results and Discussion

### 3.1. Structural, Morphological, and Elemental Characterization

The structural phases of the BOC, BOI, and BOC/BOI-3 were analyzed using XRD patterns, as shown in Figure 1a. The XRD pattern of the prepared BOC was well matched with the monoclinic characteristic of BOC (PDF#36-0760) [33]. Similarly, the XRD pattern of the prepared BOI was the same as the orthorhombic phase of BOI (PDF#040-0548) [34]. The characteristic peaks of BOC and BOI were found to co-exist in the XRD patterns of BOC/BOI-3 samples. Furthermore, the XRD pattern of BOI exhibited an intense and robust peak at 28.08°, which confirmed the preferential growth of BOI in the (312) plan. In addition, the XRD pattern of BOC exhibited a broad peak at 29.64°, which confirmed the growth of BOC in the (411) plan. These results provide evidence that the targeted phase of the composite was synthesized successfully.

The structural characteristics of the BOC, BOI, and BOC/BOI-3 were further analyzed using Raman measurements, as shown in Figure 1b. For BOC, a peak at 146.8 cm^−1^ corresponds to A_1g_ vibration originating from the stretching mode of the Bi-Cl bond [35], while the remaining peaks between 161.8 cm^−1^ to 627.3 cm^−1^ correspond to the Bi-O bond vibration [36]. For BOI, the peak at 146.8 cm^−1^ indicates E_1g_ vibration due to the internal stretching mode of Bi-I [37]. The higher-frequency peaks observed between 184.2 cm^−1^ and 591.8 cm^−1^ are related to Bi-O bonding, which are generated due to the oxygen-rich environment in BOI [38,39]. Furthermore, the presence of BOC and BOI characteristic peaks in the Raman spectra of BOC/BOI-3 verified the successful formation of a BOC/BOI heterostructure. In addition, although the positions of all observed peaks in the heterostructure are similar to the BOC and BOI samples, the peak at 506.1 cm^−1^ in the BOC/BOI-3 composite is blue-shifted and red-shifted compared to BOI and BOC, respectively. These shifts indicate that BOI and BOC in the BOC/BOI-3 composites are strongly interconnected due to the interfacial interaction between oxygen and bismuth atoms.

To investigate the morphology and microstructure of the synthesized samples, FESEM and TEM measurements were performed. Figure 2 shows SEM images of the BOC, BOI, and BOC/BOI-3 samples. The morphology of BOC was observed to be irregularly agglomerated nanoparticles of about 80 nm in size (Figure 2a,b). By contrast, SEM images of the BOI show micro-spherical structures composed of elongated nanoparticles with an approximate thickness of 100 nm (Figure 2c,d). In comparison, the SEM images of BOC/BOI-3 show micro-spherical structures composed of nano-platelets with an average thickness of 50 nm (Figure 2e,f). The nano-platelets of the BOC/BOI-3 are thinner than those of BOC and BOI.

The microstructure of the BOC/BOI heterojunction was investigated using HRTEM. Figure 3a reveals that BOC/BOI-3 composite has a 2D nano-platelets morphology with sizes varying from 50 nm to 300 nm. The HRTEM image of the yellow rectangle region marked in Figure 3a shows that 2D nano-platelets of BOC are grown on the surface of BOI (Figure 3b). A small portion (enclosed in a red rectangle in Figure 3b) is taken to obtain the IFFT pattern, as shown in Figure 3c. The intensity line profiles of the planes observed from IFFT patterns are shown in Appendix A, respectively. The fringe spacings of the perpendicular planes of BOC and BOI are determined to be 0.410 nm and 0.507 nm, respectively. According to the XRD reference file (PDF#36-0760), these measured fringe spacings correspond to the (011) and (400) planes of BOC in Figure 3c, respectively. Similarly, a small portion (enclosed in a white rectangle in Figure 3b) is taken to determine the IFFT pattern for bare BOI, as shown in Figure 3d. The intensity line profiles of these oriented planes are defined with a fringe spacing of 0.315 nm (Appendix A), which corresponds to the (312) plan of BOI according to the XRD reference file (PDF#040-0548). Altogether, the TEM analysis confirms the successful coupling of BOC with BOI, as illustrated in Figure 3e. The elemental composition of BOC/BOI-3 is confirmed by EDS mapping of the region in Appendix A, as shown in Appendix A. The sum EDS image shows the uniform distribution of the corresponding elements (Appendix A). The elemental mapping of individual Bi, O, Cl, and I clearly show that each element is uniformly distributed (Appendix A). In addition, the EDS spectrum indicates that the Bi, O, Cl, and I contents are about 44.66%, 44.25%, 4.34%, and 5.03%, respectively, as listed in the inset table of Appendix A.

Moreover, the elemental compositions of BOC, BOI and BOC/BOI-3 were further investigated using XPS. The survey XPS spectrum of the BOC/BOI-3 composite shows the presence of O 1s, I 3d, Cl 2p, and Bi 4f peaks, thus confirming the existence of BOI and BOC in the composite (Appendix A). Meanwhile, the C1s peak observed at 284.800 eV originates from the carbon tape used as a substrate for XPS measurements [40,41]. The Bi 4f XPS spectra of all samples exhibit the typical peaks of Bi 4f_5/2_ and Bi 4f_7/2_, which are attributed to the Bi-O bonding (Figure 4a). Both peaks of BOC/BOI shift to lower binding energies (0.300 eV and 0.425 eV) than BOC and higher binding energies (0.325 eV and 0.200 eV) than BOI. The O 1 s spectra of each sample were deconvoluted into two peaks, as shown in Figure 4b. The main peak at the lower binding energy and the satellite peak at the higher binding energy originate from Bi-O and adsorbed oxygen, respectively [42]. In particular, the position of the main peak in BOC/BOI is lower than that in BOC and higher than that in BOI [43], which is similar to the tendency of the Bi 4f spectra. We also compared the I 3d spectrum of BOI and the Cl 2p spectrum of BOC with the BOC/BOI composite, as depicted in Figure 4c,d. Figure 4c shows the I 3d spectra of the BOI and BOC/BOI-3 samples [44]. It can be seen that a more oxygenated environment was generated in the form of composite. Therefore, the position of both the I 3d_1/2_ and I 3d_5/2_ peaks in the composite were shifted by 0.075 eV toward a higher binding energy than BOI. The blue shift may result from a more oxygenated environment in the composite. In the Cl 2p spectrum in Figure 4d, the deconvoluted Cl 2p_1/2_ and Cl 2p_3/2_ peaks of the Cl 2p peak in the composite were also shifted by 0.300 eV and 1.000 eV toward a lower binding energy than BOC, respectively.

In XPS analysis, it is observed that all peak positions of BOC/BOI composite were shifted toward higher and lower binding energies than BOI and BOC, respectively. Since Iodine (I) is less electronegative than chlorine (Cl), it is easy for I to leave electrons. Theoretically, when I donate electrons, the concentration of the outer shell’s electron decreases, resulting in a reduced shielding effect and corresponding to an increase in the binding energy of the inner electrons. As a result, the corresponding XPS peaks shift toward a high binding energy. Therefore, this high energy shift indicates the electron density migration from BOI to BOC in the BOC/BOI-3 heterojunction. The vice versa effect is responsible for the shift of the BOC peaks to lower binding energies.

### 3.2. Photocatalytic Performance, Elucidation of Active Species, and Degradation Pathway

The photocatalytic performances of the BOC, BOI and BOC/BOI samples were evaluated by degrading RhB dye and BPA colorless pollutant in an aqueous solution under visible-light irradiation, as shown in Figure 5. Figure 5a shows that the degradation ratio (C/C_o_) of RhB dye solution over BOC/BOI composites is substantially higher than that of pure BOC and BOI. Specifically, the BOC/BOI-3 and BOC/BOI-2 composites completely decomposed RhB dye solution within 160 and 220 min, respectively. The temporal absorption spectra of RhB dye solution with BOC/BOI-3 show a gradual decrease in absorbance intensity with irradiation time (Appendix A). Eventually, the absorbance peak completely disappears after 160 min, indicating complete decolorization of RhB dye. Furthermore, during photocatalytic degradation, the position of the peak shifts from 554.5 nm to 498.5 nm, which is caused by the deethylation of the RhB dye [45].

Furthermore, the degradation rate of the RhB solution in the presence of each sample can be calculated using the pseudo-first-order kinetic model, as expressed in Equation (1),
(1)$$ln\frac{C_{0}}{C}=kt$$
where *k*, *C*_0_, and *C* represent the reaction rate constant, initial concentration, and the remaining concentration of RhB in the solution at light-exposed time *t*, respectively. We obtained the *k* of each sample via curve fitting with Equation (1) (Appendix A). The obtained reaction rate and degradation efficiency for each sample are shown in Figure 5b. BOC/BOI-3 had the highest degradation rate of 0.040 min^−1^ as well as the highest degradation efficiency of 97.68% after 90 min of light irradiation. In addition, the photocatalytic performances of BOC, BOI and BOC/BOI-3 were evaluated by decomposing the BPA in an aqueous solution under visible light (Figure 5c,d, and Appendix A). The degradation efficiency of BOC/BOI-3 over BPA was ~92.89% within 30 min, which was much higher than those of the BOC and BOI samples. The degradation rate of BOC/BOI-3 was 0.094 min^−1^, which was 120 and 6 times higher than those of bare BOC and BOI, respectively. Moreover, we compared the degradation performance of our synthesized sample with previously reported bismuth oxyhalide-based heterostructures and other bismuth-based heterostructures, as shown in Appendix A. Importantly, our grown BOC/BOI-3 sample exhibits a high degradation rate and efficiency for both RhB and BPA under visible-light illumination, compared to the reported literature.

Next, radical trapping analysis was conducted to explore the active radicals generated in the BOC/BOI-3 sample during RhB degradation, as shown in Figure 6. BQ, IPA, and KI were used as trapping agents for ^•^O_2_^−^, ^•^OH, and h^+^, respectively. Figure 6a depicts the degree of suppression of RhB degradation in the presence of BQ, IPA, and KI scavengers. The degradation of RhB in the presence of IPA is higher than those of BQ and KI. The corresponding degradation efficiency of RhB was reduced from 97% (no scavenger) to 54%, 46%, and 40% with IPA, BQ, and KI, respectively (Appendix A). Next, Appendix A shows that the degradation of RhB followed the first-order reaction kinetic model, wherein the degradation rate was reduced from 0.0400 min^−1^ (no scavenger) to 0.008, 0.007, and 0.005 min^−1^ with IPA, BQ, and KI, respectively. Thus, the results indicate that ^•^OH, ^•^O_2_^−^ radical, and h^+^ can be generated during RhB degradation [46].

To confirm the direct formation of each radical during RhB degradation, we employed NBT as a probe molecule for the detection of the ^•^O_2_^−^ radical. To directly investigate the ^•^O_2_^−^ radical, the temporal absorption spectra of NBT with UV light were obtained in the presence of the BOC, BOI, and BOC/BOI-3 samples, as shown in Appendix A. Please note that the decrease in UV absorbance at the main peak at 259 nm indicates the generation of ^•^O_2_^−^ radicals during the transformation from NBT to Formazan [47]. Compared to BOC and BOI, BOC/BOI-3 has the lowest absorbance at 259 nm after 150 min, corresponding to the highest generation of ^•^O_2_^−^ radicals (Appendix A). Based on the reaction relationship between ^•^O_2_^−^ and NBT (4:1 in molar ratio) (Appendix A), the amount of ^•^O_2_^−^ radicals can be calculated to be 37.6, 46.12, and 73.84 μmol/L after 150 min for BOC, BOI, and BOC/BOI-3, respectively (Figure 6b).

Finally, we performed LC–MS to elucidate the reaction intermediates, photocatalytic degradation pathway, and mineralization process of RhB and BPA over BOC/BOI-3 sample, as shown in Appendix A. It can be seen that the degradation products were measured at m/ɀ = 443, 391, 354, 340, 251, 194, and 130 in Appendix A. The possible degradation pathway and some relative degradation products can be proposed according to mass fragmentation results, as shown in Figure 7a. This result shows that as the photocatalytic reaction proceeds, the RhB molecules gradually divide into small molecules from m/ɀ =443 to m/ɀ =130 via N-deethylation and degradation of RhB chromophore structure. In the process of N-deethylation, the OH radicals will attack the central carbon of the RhB molecule with a high negative density charge, resulting in the degradation of RhB (m/ɀ = 443) molecules into N-deethylated intermediates first (m/ɀ = 391). Then, these N-deethylated intermediates (m/ɀ = 391) are transformed into intermediates with the m/ɀ value of 354 and 340 by OH or O_2_ radical attack on the bonding between benzene and chromophore, and further degraded into smaller m/ɀ intermediates with the m/ɀ value of 254, 251 and then to m/ɀ 194 and finally convert to m/ɀ = 130 on the cleavage of the chromophore. In Figure 7b, the systematic degradation pathway for BPA shows that the BPA molecules gradually convert into small molecules via two degradation routes. Rout (1) presents the BPA conversion into 4-n-octylphenol-mono-ethoxylate (m/ɀ: 249) by attacking active radicals on the isopropyl/aliphatic chain between the two benzene rings. In rout (2), BPA converts into (m/ɀ: 208), which may be allocated to the formation of 3 possible intermediates; (i) 1,1-di(p-toly)ethylene, (ii) 3,3-diphenylacrylaldehyde and (iii) 2-methyl-1,1diphenylpropene, and then converted into 4-isopropyle-3-methylphenol (m/ɀ: 150) by the disconnection of two aromatic rings owing to redox reaction. Thereafter, the benzene ring opens with the oxidization resulting in the breaking of C–C bonds, leading to the formation of succinic acid (m/ɀ: 118). Finally, it converts into smaller molecules, CO_2_ and H_2_O.

### 3.3. Reusability, Structural and Morphological Stability of Photocatalyst

We further conducted the recycling experiments to evaluate the photocatlytic reusability and stability of BOC/BOI-3 heterojunction photocatalyst, as illustrated in Figure 8. It can be seen that BOC/BOI-3 provides 100% RhB degradation efficiency (Figure 8a) and 95% BPA degradation efficiency (Figure 8b) after the first cycle, while a subsequent decrease in catalytic activity is observed after the second and third run. After the 3rd cycle, 85.34% of RhB and 51.64% of BPA were degraded in 160 min and 40 min, respectively. The observed decrease in efficiency is probably caused by the adsorption of organic intermediates on the catalyst surface [48]. Therefore, the structural and morphological stability of BOC/BOI-3 was verified using XRD, Raman, and SEM for the BOC/BOI-3 after the recycling experiment. In Figure 8c and Appendix A, we observed negligible variation in the XRD pattern and morphology, confirming structural and morphological stability. Additionally, the BOC/BOI-3 sample shows robust Raman intensity in a wide range because of the adsorption of the RhB molecule on the surface of BOC/BOI-3, which also reduced its photocatalytic performance in the 2nd and 3rd cycle during photocatalytic degradation (Figure 8d). However, no extra peak is observed in the Raman spectra of BOC/BOI-3 after degrading BPA due to the low concentration and less intense Raman active modes of BPA in the 125–1000 cm^−1^ range [49].

### 3.4. Behavior of Photogenerated Charge Carriers

PL measurements were conducted to investigate the recombination behavior of photo-induced e^−^-h^+^ pairs in BOC, BOI, and BOC/BOI-3, as shown in Figure 9a. The BOC/BOI-3 composite exhibited a lower PL emission intensity than BOC and BOI, which indicates the reduced recombination of e^−^-h^+^ pairs in the BOC/BOI-3 sample. Furthermore, we adapted time-resolved photoluminescence spectroscopy (TRPL) to estimate the lifetime of the excited charge carriers (Figure 9b and Appendix A). The average lifetimes (τ_avg_) of BOC/BOI-3 (8.80 ns) are longer than that of the BOC (7.42 ns) and BOI (7.91 ns). The longer τ_avg_ indicates that the recombination rate of the photogenerated charge carriers is reduced. Thus, the BOC/BOI composite led to more efficient separation of electron–hole pairs. We also measured the transient photocurrent response of the samples to examine the separation efficiency of the photo-excited charge carriers, as shown in Figure 9c. The BOC/BOI-3 has a higher photocurrent response than both the pristine BOC and BOI samples. The strong photocurrent response reveals the enhanced separation and transfer of the photogenerated charge carriers in the BOC/BOI composite. Next, an EIS Nyquist plot was taken to examine the electrode/electrolyte interfacial charge transfer resistance, as shown in Figure 9d. The Nyquist plot can be well-reproduced into solution-spreading resistance (R_s_) and charge transfer resistance (R_ct_), in parallel with the constant-phase element (CPE) (inset of Figure 9d) [50]. The R_ct_ values for BOC, BOI, and BOC/BOI-3 are 0.43, 0.64, and 0.32 MΩ, respectively. The smallest R_ct_ of BOC/BOI-3 has a lower charge transfer resistance than BOC and BOI. Therefore, these results suggest that the BOC/BOI-3 heterojunction can effectively suppress the recombination of photo-excited charge carriers, which will effectively improve the photocatalytic performance of the BOC/BOI-3 heterostructure.

### 3.5. Photocatalytic Mechanism and Interfacial Charge Transfer Behavior

To elucidate the photocatalytic mechanism of the BOC/BOI heterostructure, the electronic structures of BOC, BOI, and BOC/BOI-3 were investigated through measurements of UV–Vis diffuse reflectance spectra (DRS), Mott–Schottky (MS) plots, and valence band (VB) XPS spectra, as shown in Figure 10. First, the UV–Vis DRS of BOC, BOI, and BOC/BOI-3 were measured to investigate optical absorption ability (Figure 10a). The absorption edges for the BOC and BOI were located at around 450 nm and 425 nm, respectively. In comparison, the absorption edge of the BOC/BOI-3 was extended to a longer wavelength of 460 nm. The extended absorption edge of BOC/BOI-3 is attributed to the strong contribution of BOC in BOC/BOI-3 to the absorption of visible light. Moreover, the optical bandgap of BOC and BOI could be obtained by curve fitting of Tauc plotting of (αhν)^n^ versus E_g_ (inset of Figure 10a), where n = 1/2 for the indirect band gap semiconducting behavior of BOC and BOI [51,52]. The obtained band gaps were 2.67 eV and 3.07 eV for BOC and BOI, respectively.

Second, the Fermi energy levels (E_f_) of the prepared BOC and BOI samples were obtained using MS analysis, as shown in Figure 10b,c. The MS plots of BOC and BOI both show the positive slopes, thus indicating their n-type semiconducting behavior [53]. Furthermore, by extrapolating MS plots, the flatband potentials of BOC and BOI were obtained to be −0.645 V and −1.101 V vs. standard calomel electrode (SCE). Please note that the flatband potential (E_fb_) of the n-type semiconductor represents the E_f_ level [54]. The E_f_ vs. normal hydrogen electrode (NHE) scale were calculated using Equation (2) [55]; as a result, the E_f_ of BOC and BOI were found to be −0.401 eV and −0.857 eV vs. NHE, respectively (Figure 11a).
(2)$$E_{\mathrm{fb}}\;\left(\mathrm{vs}.\;\mathrm{NHE}\right)\;={\;E}_{\mathrm{fb}}\;\left(\mathrm{vs}.\;\mathrm{SCE}\right)\;+\;0.244\left(\mathrm{eV}\right),$$

Furthermore, valence band (VB) XPS measurement was carried out to determine the VB potential for BOC and BOI, as shown in Figure 10d. The VB XPS spectra reveal the VB maxima of 2.44 eV and 2.60 eV for BOC and BOI, respectively. Thus, the VB potentials with respect to E_f_ level are calculated to be 2.039 V and 1.743 V vs. NHE for BOC and BOI, respectively (Figure 11a).

Furthermore, the CB minima of BOC and BOI were determined using the following Equation (3).
(3)$$E_{\mathrm{CB}}\;\left(\mathrm{vs}.\;\mathrm{NHE}\right)\;=\;E_{\mathrm{VB}}\;\left(\mathrm{vs}.\;\mathrm{NHE}\right)\;-\;E_{g},$$
where E_CB_, E_VB_, and E_g_, denote the CB potential, VB potential and band gap of the sample, respectively. As a result, the calculated CB potentials for BOC and BOI are −0.631 V and 1.327 V vs. NHE, respectively (Figure 11a). Thus, based on the above analysis, the BOC/BOI will make a type-II heterojunction. When BOC and BOI formed the BOC/BOI heterojunction, the electrons would spontaneously migrate from BOI to BOC through the BOC/BOI interface to align the Fermi level because BOI has a higher E_f_ than BOC. The migration behavior of the charge carriers at the interface is also consistent with the elemental analysis in XPS spectra (as discussed in Section 3.1). This electron migration generates a depletion region at the BOC and BOI interface with an internal electric field (IEF) oriented from BOI to BOC, therefore resulting in the band bending upward for BOI and downward for BOC, respectively, as shown in Figure 11b.

Regarding the analysis of the band structure of the BOC/BOI-3 sample, a possible Z-scheme photocatalytic mechanism for BOC/BOI could be proposed (Figure 11c). Under visible-light irradiation, the charge carriers are excited simultaneously from the VB to the CB in both BOC and BOI to produce photogenerated e^−^ and h^+^ (as shown in the below reactions (4) and (5)). Then, the photo-excited electrons in the CB of BOC will be easily transferred to the VB of BOI due to the IEF at the interface in the heterojunction; this initiates the Z-scheme charge transport of the photo-excited charge carriers in the photocatalytic processes (reaction (6) and in Figure 10c). As a result, the remaining photogenerated holes in the VB of BOC with strong oxidation ability will split H_2_O into ^•^OH and hydrogen ions (H^+^) (reaction (7)) due to the fact that the VB of BOC (2.039 eV) is more positive than +1.99 eV (H_2_O/^•^OH vs. NHE). At the same time, the remaining photogenerated electrons in the CB of BOI will reduce the O_2_ molecules in aqueous solution into highly active ^•^O_2_^−^ radical (reaction (8)), then the ^•^O_2_^−^ and ^•^OH^−^ radicals further react with RhB or BPA aqueous solution to degrade it into H_2_O, CO_2_, and small molecules (reaction (9)). Moreover, the unique 3D morphology of BOC/BOI-3 will provide more catalytic centers and increase the active sites. Accordingly, BOC/BOI-3 shows enhanced photocatalytic performance for degrading RhB and BPA pollutants.
(4)$$\mathrm{BOC}+h\nu \;\to {\mathrm{BOC}}^{*}\;\left(e^{-}+h^{+}\right),$$
(5)$$\mathrm{BOI}+h\nu \;\to {\mathrm{BOI}}^{*}\;\left(e^{-}+h^{+}\right),$$
(6)$${\mathrm{BOC}}^{*}\left(e^{-}+h^{+}\right)+\mathrm{BOI}\;\left(e^{-}+h^{+}\right)\to {\mathrm{BOC}}^{*}\;\left(h^{+}\right)+{\;\mathrm{BOI}}^{*}\;\left(e^{-}\right),$$
(7)$$H_{2}O+h^{+}\overset{+1.99\;V}{\to }\;{}^{•}\mathrm{OH}+H^{+},$$
(8)$$O_{2}+e^{-}\;\overset{-0.33\;V}{\to }\;{}^{•}O_{2}^{-},$$
(9)$$\mathrm{RhB}/\mathrm{BPA}+{}^{•}\mathrm{OH}/{}^{•}O_{2}^{-}\to {\;H}_{2}O+{\mathrm{CO}}_{2}+\mathrm{other}\;\mathrm{decomposed}\;\mathrm{products},$$

## 4. Conclusions

In this study, a simple solvothermal growth technique was used to synthesize the 3D hierarchical BOC/BOI heterostructure photocatalyst. The heterostructure demonstrated direct Z-scheme photocatalytic behavior in the photocatalytic degradation of RhB and BPA. The 3D hierarchal morphology of BOC/BOI can increase active sites due to its high surface area, and can improve visible-light absorption via multiple reflections inside the sample. XPS and mechanistic studies proved the generation of an internal electric field from BOI to BOC at their interface, which contributed to Z-scheme transport of photogenerated charge carriers in the BOC/BOI heterojunction. Thus, the BOC/BOI sample demonstrated a higher reaction rate and a higher degradation efficiency than the bare BOC and BOI during RhB and BPA degradation. Radical trapping analysis showed that ^•^O_2_^−^, and ^•^OH radicals played a crucial role in RhB and BPA degradation. Through LC–MS results of RhB and BPA during photodegradation, its mineralization and degradation pathway were proposed. Therefore, the excellent photocatalytic performances of BOC/BOI are attributed to its large surface area, visible-light absorption, and low electron–hole recombination, which are due to its unique 3D hierarchical morphology and the direct Z-scheme photocatalytic process. Conclusively, direct Z-scheme BOC/BOI heterostructure is a suitable photocatalyst for use in the photodegradation of environmental pollutants.

## Figures and Tables

**Figure 1 nanomaterials-12-00767-f001:**
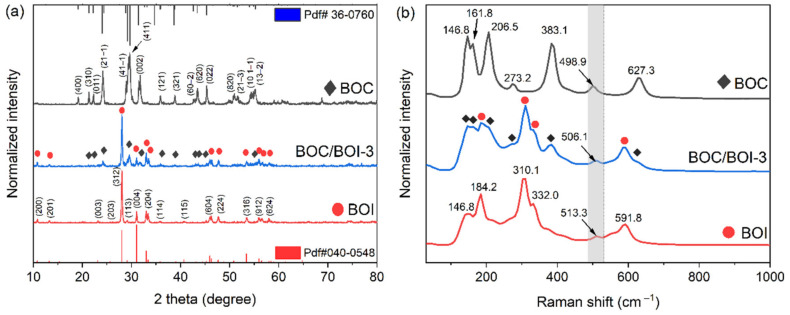
(**a**) XRD patterns and (**b**) Raman spectra of BOC, BOI, and BOC/BOI-3 heterostructure photocatalysts.

**Figure 2 nanomaterials-12-00767-f002:**
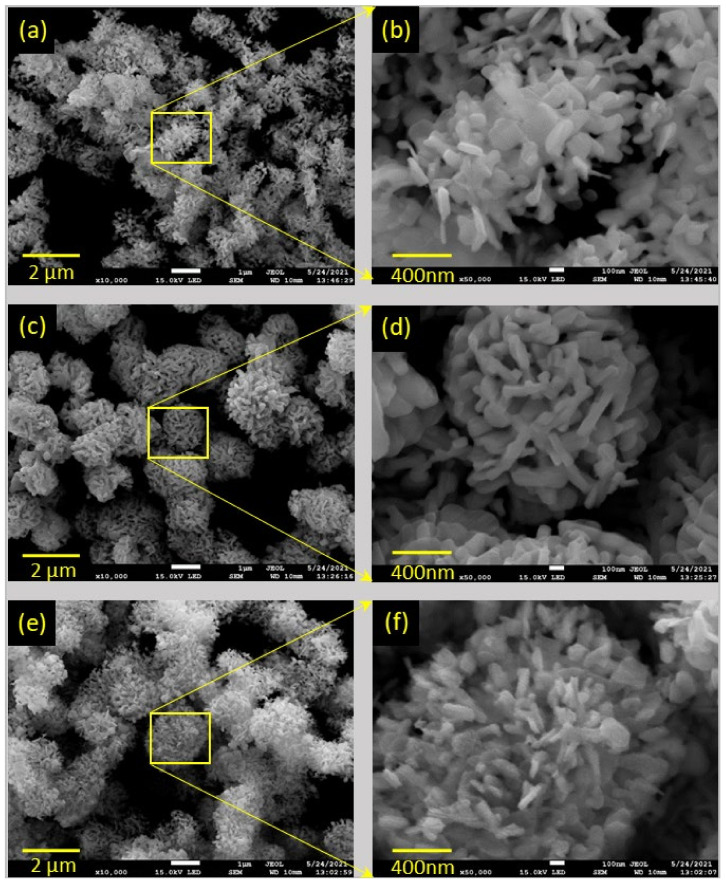
FESEM images of (**a**,**b**) BOC, (**c**,**d**) BOI, and (**e**,**f**) BOC/BOI-3 heterostructures.

**Figure 3 nanomaterials-12-00767-f003:**
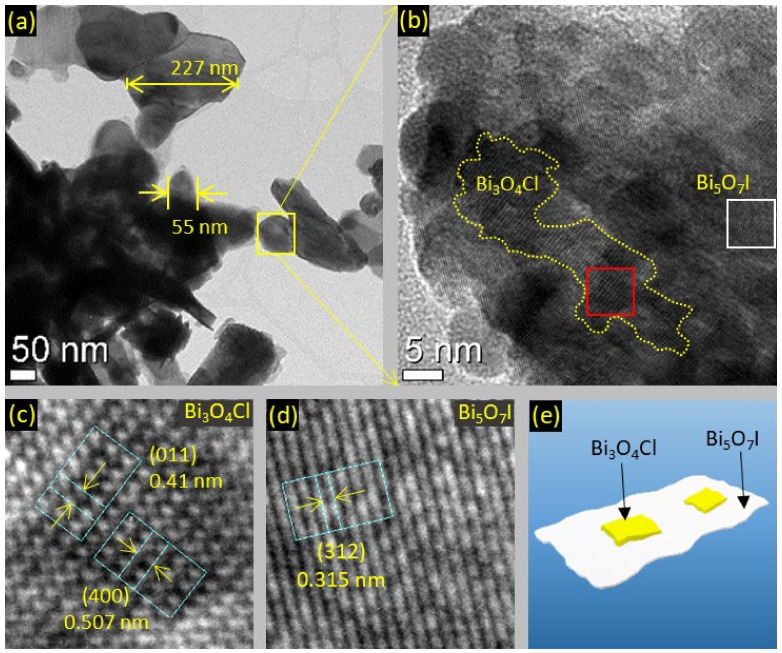
HRTEM images of (**a**,**b**) BOC/BOI-3 nano-platelets, (**c**) IFFT pattern of BOC and (**d**) BOI in BOC/BOI, and (**e**) schematic of BOC/BOI-3 heterostructure.

**Figure 4 nanomaterials-12-00767-f004:**
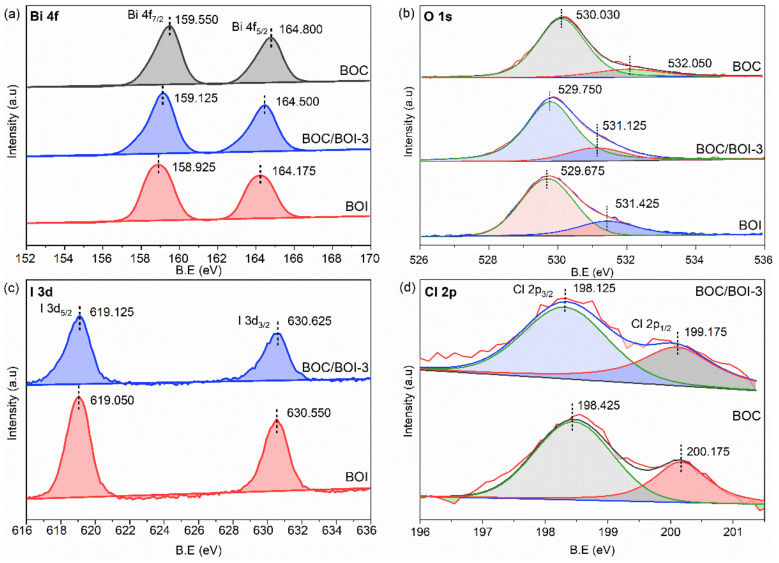
(**a**) High-resolution XPS core level data for Bi 4f, (**b**) O 1s, (**c**) I 3d, and (**d**) Cl 2p obtained from BOC, BOI, and BOC/BOI-3 samples.

**Figure 5 nanomaterials-12-00767-f005:**
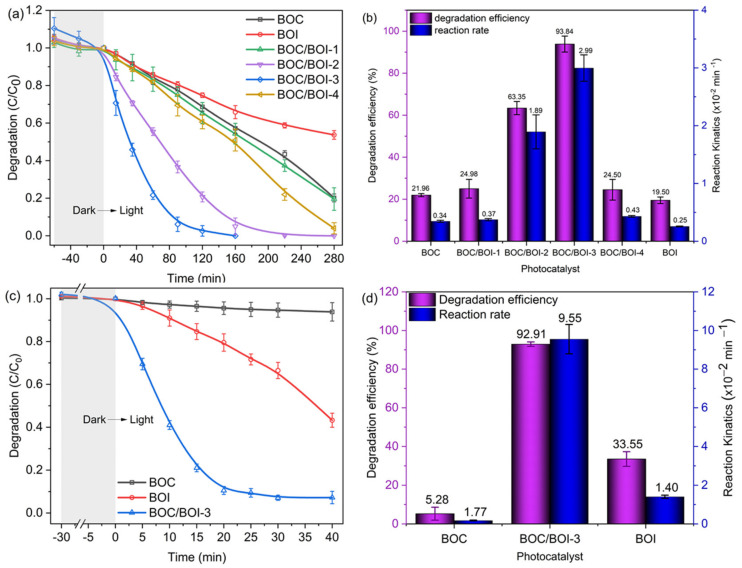
(**a**) Photocatalytic degradation of RhB and (**b**) degradation efficiency of RhB with the reaction rate constants employed by all samples; (**c**) Photocatalytic degradation of BPA and (**d**) its degradation efficiency with reaction rate constants employed by BOI, BOC and BOC/BOI-3 samples.

**Figure 6 nanomaterials-12-00767-f006:**
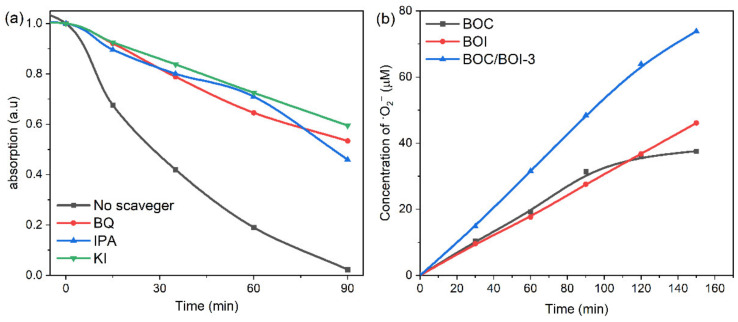
(**a**) Radical trapping experiments during degradation of RhB over BOC/BOI-3, (**b**) time-dependent concentration plots of the amount of ^•^O_2_^−^ radicals.

**Figure 7 nanomaterials-12-00767-f007:**
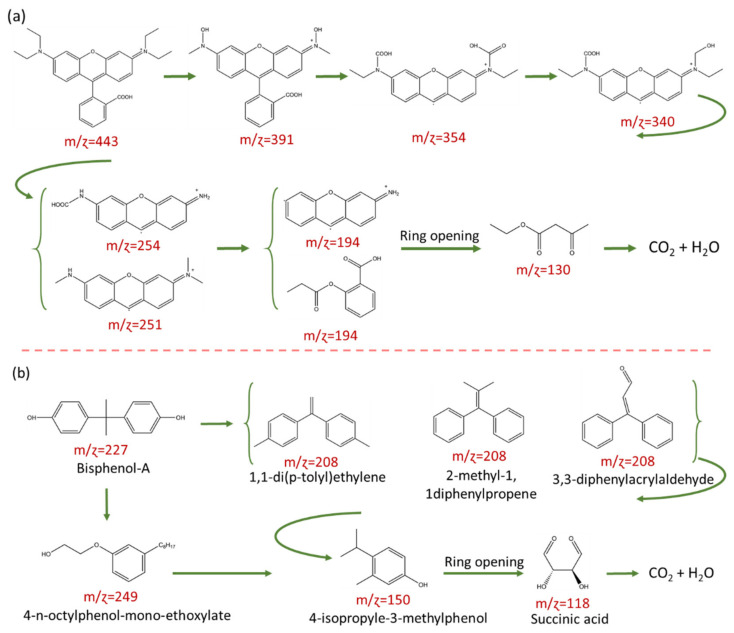
Photocatalytic degradation pathway and possible reaction intermediates in (**a**) RhB and (**b**) BPA over BOC/BOI-3.

**Figure 8 nanomaterials-12-00767-f008:**
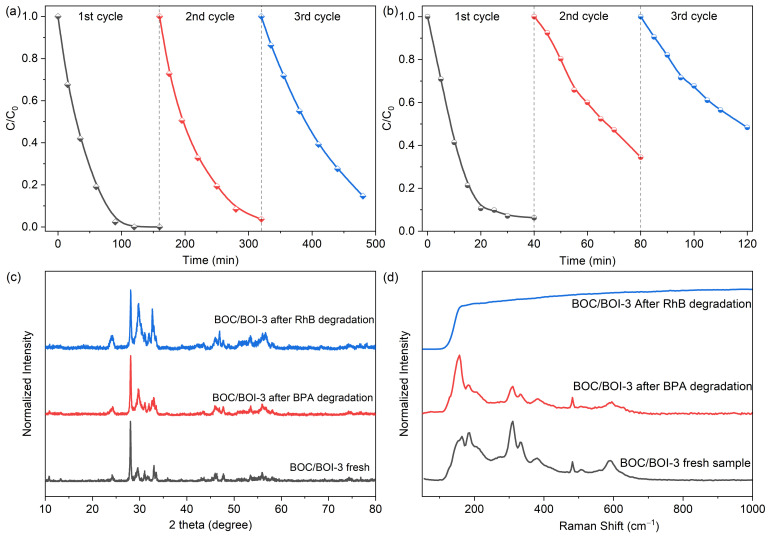
(**a**) Cyclic degradation test of Rhodamine-B and (**b**) Bisphenol-A over BOC/BOI-3 sample, (**c**) XRD pattern and (**d**) Raman spectra of BOC/BOI-3 sample before and after BPA and RhB degradation.

**Figure 9 nanomaterials-12-00767-f009:**
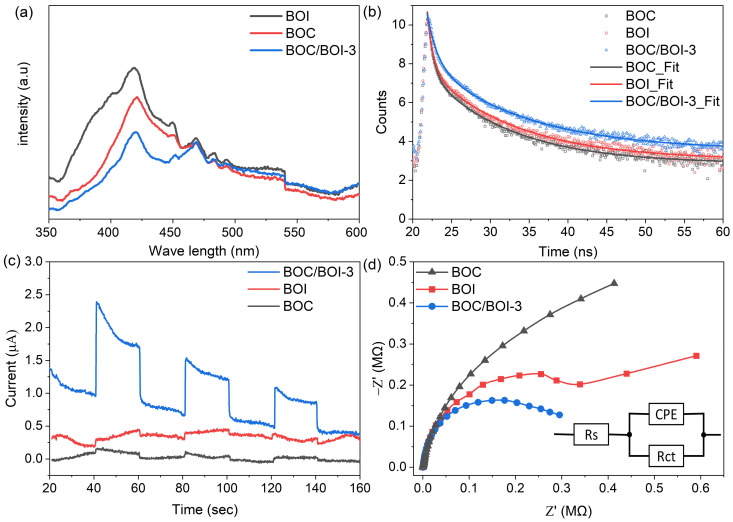
(**a**) Photoluminescence spectra, (**b**) TRPL decay spectra, (**c**), transient photocurrent response and (**d**) EIS Nyquist plots of BOC, BOI, and BOC/BOI-3 samples.

**Figure 10 nanomaterials-12-00767-f010:**
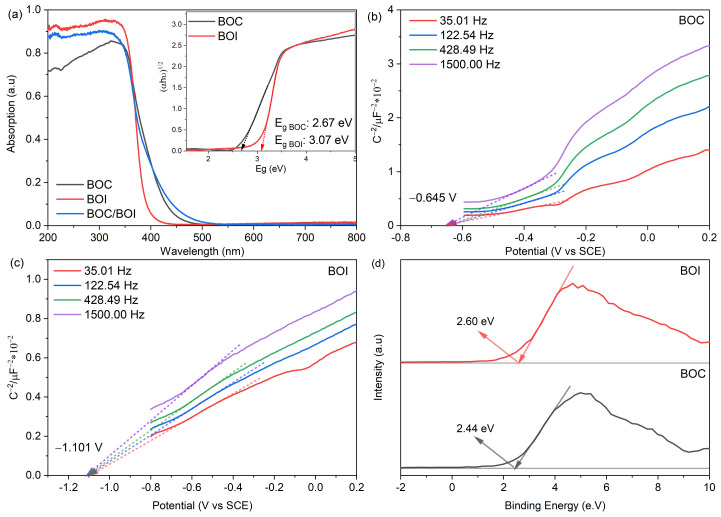
(**a**) UV–Vis diffuse reflectance spectra of BOC, BOI and BOC/BOI-3, (inset) showing Tauc plot of BOC and BOI, (**b**,**c**) Mott–Schottky plot of BOC and BOI, (**d**) Valence band XPS of BOC and BOI.

**Figure 11 nanomaterials-12-00767-f011:**
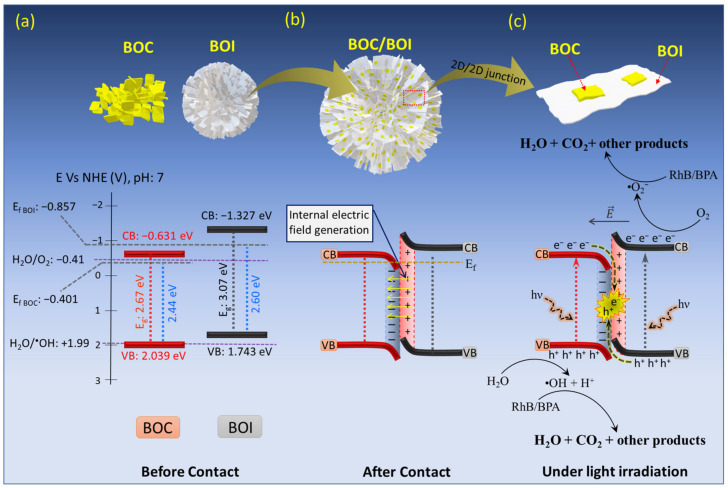
Schematics and energy-level diagrams of (**a**) BOC and BOI before contact, (**b**) IEF formation at BOC and BOI interface after contact, and (**c**) interfacial charge transfer behavior in BOC/BOI-3 heterostructure under visible-light irradiation.

## Data Availability

Not applicable.

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
