# Peer review of "Z-Scheme Heterojunction of 3-Dimensional Hierarchical Bi3O4Cl/Bi5O7I for a Significant Enhancement in the Photocatalytic Degradation of Organic Pollutants (RhB and BPA)"

_nanomaterials, 2022, doi:10.3390/nano12050767_

Round 1
Reviewer 1 Report
A brief summary
The 3D hierarchical Bi3O4Cl/Bi5O7I heterostructure was proposed for the photocatalytic destruction of organic pollutants (Rhodamine-B and Bisphenol-A) in an aqueous medium under visible light. The •O2- , and •OH radicals generated in the photocatalyst played a crucial role in degradation of these organic pollutants. The excellent photocatalytic performances of this photocatalyst are due to its large surface area, visible light absorption, and low electron-hole recombination. This photocatalyst is proposed for wastewater treatment from organic impurities.
Broad comments
A well-written paper on the current topic of finding an effective and cheap way to treat wastewater from organic pollutants. Complete information on the production of photocatalyst, its properties and mechanism of action is presented.
Specific comments
No specific comments - no editing required.
Author Response
please sse the attahchment.

Reviewer 2 Report
Yang et al. describes the use of Z-scheme heterojunction photocatalyst in the photocatalytic degradation of rhodamine-B and bisphenol-A from water. The research topics discussed by the authors are important from the point of view of the problems related to the removal of industrial pollutants from water. I have no substantive objections to the presented manuscript. The results are clearly presented and supported by numerous measurement techniques. My minor remarks relate only to the editorial aspect:
1) In the formula of bismuth nitrate pentahydrate the dot should be in the middle between the nitrate and water formulas (Page 3, lines 123, 127 and further in the text.
2) Lack of the subscript in the formula (Page 3, line 127).
3) 'N-de-deethylation' - is it correct? (Page 10, line 375).
4) '3,3-diphenylacryladehide' (Page 10, line 387).
5) Replace dot with comma ((...)cycle, while(...) (Page 11, line 401).
Having regard to the above editorial corrections, I can recommend this manuscript for publication in Nanomaterials.
Reviewer 3 Report
In this paper, the authors report the synthesis of three-dimensional hierarchical BOC / boi heterostructures, which photocatalytic degrade RHB dyes and col- or phenol free bisphenol A (BPA) pollutants under visible light. the highly improved photocatalytic performance of the BOC/BOI heterostructure is attributed to the synergetic effects of large surface area, high visible light absorption, and the enhanced separation and transport of photoexcited electron-hole pairs induced by the hierarchical and Z-scheme heterojunction of the BOC/BOI. There are some issues which the authors should address them before acceptance process of the paper. Here are my comments:
- The clarity and text of some pictures are too small, and the author needs to make some adjustments. For example, figure 1, 5, 7, 8, 10, 11.
- In figure 2e-f, how can BOI and BOC be distinguished?
- For high efficiency photocatalysts, the following work should be mentioned by the author, such as: J. Catal. 373 (2019)161-172; Bull. Korean Chem. Soc. 34 (2013) 3039–3045; Appl. Catal. A-Gen. 590 (2020) 117342; Appl. Catal. B-Environ. 248 (2019) 380-387; Org. Biomol. Chem., 2018,16, 2406-2410; Electrochimica Acta 216(2016) 517-527; Colloids and Surfaces A: Physicochemical and Engineering Aspects 633(2)(2022) 127918
Round 2
Reviewer 3 Report
The article has been systematically modified and can be accepted.